# Characterization of Human Immunodeficiency Virus-1 Transmission Clusters and Transmitted Drug-Resistant Mutations in Croatia from 2019 to 2022

**DOI:** 10.3390/v15122408

**Published:** 2023-12-11

**Authors:** Ana Planinić, Josip Begovac, Filip Rokić, Petra Šimičić, Maja Oroz, Katja Jakovac, Oliver Vugrek, Snjezana Zidovec-Lepej

**Affiliations:** 1Department of Immunological and Molecular Diagnostics, University Hospital for Infectious Diseases Dr. Fran Mihaljević, 10000 Zagreb, Croatia; anaplaninic@yahoo.com; 2School of Medicine, University of Zagreb, 10000 Zagreb, Croatia; josip.begovac@gmail.com; 3Ruđer Bošković Institute, 10000 Zagreb, Croatia; frokic@irb.hr (F.R.); katjajakovac@gmail.com (K.J.); oliver.vugrek@irb.hr (O.V.); 4Department of Oncology and Nuclear Medicine, Sestre Milosrdnice University Hospital Center, 10000 Zagreb, Croatia; petrasimicic@gmail.com; 5Cytogenetic Laboratory, Department of Obstetrics and Gynecology, Clinical Hospital Sveti Duh, 10000 Zagreb, Croatia; majatrupkovic@gmail.com

**Keywords:** epidemiology, transmitted drug resistance, surveillance drug resistance mutations, phylogenetics

## Abstract

Molecular epidemiology of HIV-1 infection is challenging due to the highly diverse HIV-genome. We investigated the genetic diversity and prevalence of transmitted drug resistance (TDR) followed by phylogenetic analysis in 270 HIV-1 infected, treatment-naïve individuals from Croatia in the period 2019–2022. The results of this research confirmed a high overall prevalence of TDR of 16.7%. Resistance to nucleoside reverse transcriptase inhibitors (NRTIs), non-nucleoside RTIs (NNRTIs), and protease inhibitors (PIs) was found in 9.6%, 7.4%, and 1.5% of persons, respectively. No resistance to integrase strand-transfer inhibitors (INSTIs) was found. Phylogenetic analysis revealed that 173/229 sequences (75.5%) were part of transmission clusters, and the largest identified was T215S, consisting of 45 sequences. Forward transmission was confirmed in several clusters. We compared deep sequencing (DS) with Sanger sequencing (SS) on 60 randomly selected samples and identified additional surveillance drug resistance mutations (SDRMs) in 49 of them. Our data highlight the need for baseline resistance testing in treatment-naïve persons. Although no major INSTIs were found, monitoring of SDRMs to INSTIs should be continued due to the extensive use of first- and second-generation INSTIs.

## 1. Introduction

For efficient monitoring of the HIV-1 epidemic and optimizing first-line therapy for the management of HIV-infection, the evaluation of drug resistance, surveillance drug resistance mutations (SDRMs), and subtype diversity, and the identification of new recombinant forms remain important tools of public health strategies [1,2]. With the introduction of new antiretroviral treatment options and expanded use of pre-exposure prophylaxis (PrEP), monitoring of drug resistance patterns is crucial to preserve an efficient virological response to current antiretroviral therapy (ART) as well as to future treatment options [3,4,5].

HIV-1 is characterized by extensive heterogeneity, which has a significant impact on, among other things, clinical management of HIV infection [2]. HIV-1 is classified into four groups (groups M, N, O, and P) with group M being responsible for most infections globally [6,7]. Group M is further divided into nine genetically distinct subtypes (A–D, F–H, J, K, and L), eight sub-subtypes (A1–A6, F1, F2), and, to date, 140 circulating recombinant forms (CRF) [8]. Globally, subtype C is the most prevalent and is linked to sub-Saharan African and Indian populations, followed by subtype A (east Africa) and subtype B (western Europe, United States, and Australia). Other major subtypes (F, H, J, and K) that remained stable are less prevalent, accounting for around 1% of infections worldwide, whilst subtype D has decreased over time [9,10].

Due to population mixing, changes in geographical patterns in the HIV subtype distribution were observed in recent years [11]. Increasing prevalences of non-B subtype and CRFs were reported in Europe, Australia, and the USA [11,12,13,14], specifically subtype G and CEF01_AE, which increased from 5.0% in 2008 to 8.5% in 2016 in the USA [12]. These observed changes in the epidemiology of HIV may consequently have a negative effect on resistance pathways and HIV progression, so the close surveillance and early detection of transmitted drug resistance (TDR) in ART-naïve individuals remains a focus of the WHO 2017 guidelines [15].

Although ART has been crucial for decreasing morbidity and mortality in people living with HIV (PLHIV), the emergence of HIV resistance can compromise its effectiveness [1,2]. Transmitted HIV drug resistance mutations (DRMs) in newly infected persons can have a negative impact on the efficiency of first-line antiretroviral therapy [16], and therefore all of the major guidelines recommend HIV drug-resistance testing before starting ART [17,18].

Croatia is a southeastern European country with low-level HIV epidemics (the cumulative number of infections in the period 1986–2022 was 2017), a centralized system of diagnostics and care, as well as universal free access to ART [19]. All persons infected with HIV are treated at the University Hospital for Infectious Diseases (UHID) in Zagreb, allowing us to perform extensive studies since all data are available in one place [20].

The first cases of HIV-1 subtype B infections in Croatia were observed in labour migrants who returned from western Europe, while non-B infections and CRFs were observed in seafarers who acquired HIV in Africa and eastern Asia [20].

Molecular analysis of HIV subtypes in the period 2001–2003 showed a high prevalence of subtype B (>74%) with non-B subtypes found only in heterosexuals [21]. A study on primary resistance in treatment-naïve HIV-infected patients conducted from 2006–2008 showed a high prevalence of subtype B (89%) among men who have sex with men (MSM), with only 11% of patients infected with non-B subtypes [22].

The Croatian HIV epidemic has previously been characterized by a high prevalence of TDR despite it being a country with low-level epidemics [22]. A national study on TDR in the period 2006–2008 showed a prevalence of 22%, which is in contrast to an overall prevalence of 8.3% identified in 25 European countries and Israel as a part of the European SPREAD study (study period 2008–2010) [22,23]. In comparison to other countries in the region, the differences are as follows. Countries like Serbia, Slovenia, Bulgaria, and Greece reported a substantially lower TDR when compared to Croatian data (8.8%, 2.4%, 5.2%, and 6%, respectively), while data from Hungary, Romania, and northern Italy are more comparable to Croatian settings (17%, 14.8%, and 12%, respectively) [24,25,26,27,28,29,30].

The last national study on TDR conducted before the COVID-19 pandemic estimated a prevalence of 16.4%. That study included 403 persons who entered clinical care in the period 2014–2017 [31]. The majority of patients had mutations conferring resistance to only one class of drugs. Resistance to nucleoside reverse transcriptase inhibitors (NRTIs), non-nucleoside RTI (NNRTIs), and protease inhibitors (PIs) was found in 11.4% (46/403), 6.7% (27/403), and 2.5% (10/403) of persons, respectively, with the addition of a single case (1.0%, 1/100) of resistance to integrase strand-transfer inhibitors (INSTIs). In nine individuals (2.2%), triple class resistance was detected. Phylogenetic analysis showed that 86.1% of sequences were a part of transmission clusters and identified forward transmission resistance. The majority of infected people were carrying subtype B (91.3%, 368/403). The largest TDR cluster with mutation T215S was shown to have originated from 1992 [31].

In the last two decades, there was a shift in transmission routes in Croatia, and the majority of HIV infections in the country are now acquired through MSM contact, in contrast to 1985–2010 when 40–50% of new HIV-1 cases were attributed to MSM transmission [22,31,32,33,34].

In this study, we reported the most recent data on the molecular epidemiology of HIV-1 infection and TDR in Croatia to explore the current prevalence and patterns of TDR among ART-naïve HIV-infected individuals during the period 2019–2022. We used Sanger sequencing (SS) as well as deep sequencing (DS) for analysing the prevalence of TDR and the HIV-1 diversity, and we used a phylogenetic approach to characterize HIV transmission. We have performed an extensive analysis of sequences of more than 90% of treatment-naïve persons who entered clinical care at UHID to determine the HIV-1 subtype distribution and phylogenetic structure within Croatia from 2019 to 2022, with the aim of monitoring HIV-1 infections and its transmission networks.

## 2. Materials and Methods

### 2.1. Study Population

This study was conducted at the Department for Immunological and Molecular Diagnostics at UHID. Data collection and analysis were performed on HIV-1 RNA plasma samples collected from newly diagnosed HIV patients over a four-year period during the diagnostic process and clinical follow-up at UHID. All newly diagnosed treatment-naïve PLHIV over 18 years who entered clinical care at UHID in the period between January 2019 and December 2022 with plasma viraemia > 1000 HIV-1 RNA copies per ml were included in this study. A total of 270 persons met the inclusion criteria. Further epidemiological and demographic data on the patients were supplied by the Reference Centre for HIV/AIDS.

### 2.2. RNA Extraction, Amplification, and Sequencing

At least 500 μL of the patient’s plasma samples were concentrated at 20,000 g and 8 °C for 1 h. A portion of the supernatant was removed, and the remaining 140 μL was used for RNA extraction using a QIAamp Viral RNA Mini Kit (Qiagen, Hilden, Germany) according to the manufacturer’s protocol.

Amplification of the HIV-1 *pol* gene was performed for two regions in two separate reactions: (1) sequencing of the HIV-1 protease and partial sequencing of the reverse transcriptase gene; and (2) sequencing of the HIV-1 integrase gene.

The first round of amplification was performed by using the SuperScript III One-Step RT-PCR System with Platinum *Taq* (Invitrogen, Carlsbad, CA, USA) and a region-specific primer set while nested PCR was carried out using AmpliTaq Gold™ DNA Polymerase with Gold Buffer and MgCl_2_ (Applied Biosystems, Foster City, CA, USA) with an inner primer set. Nested PCR amplicons were purified and sequenced with a BigDye Terminator v3.1 Cycle Sequencing Kit (Thermo Fisher Scientific, Waltham, MA, USA) with a set of four primers for both sequencing regions to obtain bidirectional sequences (Appendix A).

### 2.3. Genotyping and Drug Resistance Mutations Analysis

Sequences were aligned and compared with the reference strain HIV-1 HXB2 (GenBank number K03455) by using SnapGene software 7.0 (www.snapgene.com (accessed on January–October 2023).

Primary resistance to antiretroviral drugs was defined as the presence of ≥1 mutation placed on the WHO surveillance for drug resistance mutations (SDRM) list. Clinically relevant resistance to NRTIs, NNRTIs, PIs, and INSTIs was evaluated with the Stanford University HIV Drug Resistance Database, Genotypic Resistance Interpretation Algorithm version 9.5 [35] and the IAS Drug Resistance Mutation list [36]. A Stanford score of 15 to 29 predicts low-level resistance and a score of 60 or above predicts high-level resistance [35].

HIV-1 strains were defined as resistant if carrying at least one SDRM. The overall prevalence was defined as the percentage of patients with SDRMs. The prevalence of TDR for the different drug classes (NRTI, NNRTI, PI, and INSTI), was defined in the same matter.

HIV-1 subtypes and circulating recombinant forms (CRFs) were determined using the Rega HIV-1 Subtyping Tool version 3.0., COntext-based Modelling for Expeditious Typing (COMET), and the jumping profile Hidden Markov Model (jpHMM) [37,38,39].

### 2.4. Deep Sequencing Analysis

A total of 60 samples that were previously sequenced by SS were randomly selected and subjected to NGS. Previously extracted RNA was reverse transcribed with SuperScript III First-Strand Synthesis System for RT-PCR (Invitrogen) and UNINEF primers [40]. Precisely, the whole HIV-1 protease region, part of the reverse transcriptase region (K03455 number for the gene-specific position 2189–3753), and the integrase region (K03455 number for the gene-specific position 4180–5200) were sequenced on the Illumina platform using MiniSeq (Illumina, San Diego, CA, USA). Four separate multiplex PCR reactions using ALLin Taq DNA Polymerase (highQu, Kraichtal, Germany) were made for each sample with 22 primer pairs that span the target region (Appendix A). Amplicons for each sample were then pooled and purified with Agencourt Ampure XP beads (Beckman Coulter, Krefeld, Germany). Sequencing libraries were prepared using the NEBNext Ultra II DNA Library Prep Kit for Illumina (New England BioLabs, Beverly, MA, USA), according to the manufacturer’s protocol. The library concentrations and purification were measured with an Agilent High Sensitivity Kit (Agilent Technologies, Santa Clara, CA, USA) on Bioanalyzer 2100. Libraries were sequenced on MiniSeq using the MID output 300 cycles reagent kit (paired-end; 151 + 151). Raw reads were trimmed using the cutPrimer tool to remove the synthetic primer sequences used in the multiplex PCR reactions. Trimmed data in the form of FASTAQ files were analysed using the HyDRA Web tool (http://hydra.canada.ca/ (accessed on July–October 2023) according to the HyDRA Web User Guide [41]. Data were analysed with the following settings: sensitivity threshold (5%), default target coverage (10,000 reads), default filtering settings (length cut-off: 30; score cut-off: 30; minimum variant quality: 30; minimum read depth: 50) [41]. The frequencies of DRMs were interpreted using the HIVdb algorithm version 9.5 [35].

### 2.5. Phylogenetic Analysis

A total of 229/270 (84.8%) PR and RT sequences were subjected to phylogenetic analysis. Phylogenetic trees were constructed for the most frequent subtypes so the final dataset consisted of 191 subtype B sequences, 28 BD recombinant sequences, and 10 subtype A1 sequences. The dataset was supplemented with ten sequences of high similarity for each sample sequence by using the BLAST (Basic Local Alignment Search Tool) database [42]. The ElimDupes tool available at the Los Alamos database (https://www.hiv.lanl.gov/ (accessed on July–October 2023) was used to remove duplicate sequences [8].

Multiple sequence alignment for all datasets was performed in ClustalX v.2.1 and edited in AliView v.1.27 [43,44].

To rule out clustering bias due to the presence of drug resistance mutations (DRM), phylogenetic trees were constructed with the exclusion of 43 codons that are DRM sites, resulting in a final sequence size of 858 bp. The following major drug resistance mutations, according to the IAS list, were removed: PR: 23, 24, 30, 32, 46, 47, 48, 50, 53, 54, 73, 76, 82, 83, 84, 85, 88, 90; RT: 41, 65, 67, 69, 70, 74, 75, 77, 100, 101, 103, 106, 115, 116, 151, 179, 181, 184, 188, 190, 210, 215, 219, 225, 230.

Maximum likelihood trees for all dataset groups were constructed in *Mega* v.10.2.6 [45] under 1000 bootstrap replicates with the best-fitting nucleotide substitution model (GTR+I+G) for each dataset selected according to Bayesian Information Criterion (BIC). Two sequences of subtype A1 were used to root the subtype B (accession numbers OR606275 and OR606276) and the B-D recombinant (accession numbers OR606048 and OR613874) trees, and two sequences of subtype B (accession numbers OR606211 and OR606184) were used to root the subtype A1 tree.

Transmission clusters (TCs) were identified in the maximum likelihood trees using *ClusterPicker* v.1.2.3. [46] with a genetic distance and bootstrap support threshold of 4.5% and 90%, respectively. Clusters were defined as a group of three or more persons/sequences on the same branch of the phylogenetic tree. Local TCs were defined as clusters consisting of >75% of Croatian sequences, while mixed TCs were defined as clusters consisting of <75% of Croatian sequences. The final review and arrangement of the obtained phylogenetic trees was made in iTOL v.6.4.2 [47].

### 2.6. Statistical Analysis

Descriptive statistics were used to summarize the main features of the data. Values were described by percentages and frequencies or median with first and third quartiles (Q1, Q3). The association between the selected variables was analysed by Pearson’s chi-squared test or Fisher’s exact test for categorical data and by the Mann–Whitney test for continuous data. Statistical significance was defined as *p* < 0.05 two-sided. All data were analysed using the statistical R software version 2.13.1 (SAS Institute, Cary, NC, USA).

## 3. Results

### 3.1. Study Population

A total of 280 newly diagnosed treatment-naïve persons entered the Croatian Reference Centre for HIV/AIDS (86 in 2019, 58 in 2020, 68 in 2021, and 68 in 2022) from January 2019 to December 2022. Ten people did not meet the inclusion criteria, so the final data set consisted of 270 persons, providing coverage of 96.4%. The study group consisted primarily of men (254/270, 94.1%) with only 16/270 (5.9%) women making it to the final analysis. The median age at HIV diagnosis was 38 years and the main acquisition risk was MSM (231/270, 85.6%) followed by heterosexual contact (32/270, 11.9%). At the time of genotypic resistance testing, the median CD4 T cell count was 284 cells/μL (IQR 95–460) and the median plasma HIV-1 RNA level was 5.0log10 copies/mL (IQR 4.4–5.5). Characteristics of newly diagnosed PLHIV in the period 2019–2022 are shown in Table 1.

More than eight different HIV-1 genotypes and CRFs were identified within the cohort, with subtype B being the most common (73.3%, 198/270), followed by subtype A1 (4.4%, 12/270); the other genotypes were CRF 12_BF (1.5%, 4/270), CRF01_AE (1.5%, 4/270), subtype C (1.1%, 3/270), subtype F1 (1.1%, 3/270), and CRF06_CPX (0.7%, 2/270). A significant number of sequences were defined as recombinants, and according to the REGA Subtyping Tool, the most frequent were B-D (10.7%, 29/270), followed by A1-C (1.5%, 4/270) and F1-B (1.1%, 3/270) recombinants (Appendix A).

### 3.2. Prevalence of TDR and SDRMs

HIV drug resistance analysis was performed on the viral sequences of 270 persons covering the RT/PR and IN regions. The IN region for six people failed to amplify.

The overall prevalence of SDRMs was estimated at 16.7% (45/270) (Appendix A).

A total of 42/270 (15.6%) persons were identified with one drug class mutation, one person had mutations associated with resistance to two drug classes, and two individuals harboured triple class resistant virus (SDRM NRTI + NNRTI + PI).

Resistance to NRTI was the most common (9.6%; 26/270), followed by NNRTI (7.4%; 20/270) and PI (1.9%; 5/270).

Among the NRTI mutations, T215S was the most frequent (8.1%, 22/270) followed by L210W (1.5%, 4/270). Mutations K103N (3.7%, 10/270) and K101E (3.3%, 9/270) were the most frequent NNRTI mutations followed by L100I (1.1%, 3/270). PI mutations V32I, I47V, and I85V were each detected in two samples (0.7%). No major INSTI mutations were identified; however, the integrase resistance associated accessory mutation L74I was found in 7 (2.6%) integrase sequences, of which 5 were classified as non-B subtypes.

All identified SDRMs were clinically relevant according to the International AIDS Society (IAS) list 2022, except T215S, T215D, T215E, K219R, and T69D. Mutation I85V was not listed as a major or accessory DRM on the Stanford drug resistance mutation list. One individual harboured non-polymorphic accessory mutation A98G, and although it is not included on the WHO SDRM list, it is clinically relevant. The estimated prevalence of TDR to the selected antiretrovirals according to the Stanford scoring system is shown in Appendix A.

Mutations E138A and E138K, which are clinically relevant according to the IAS list 2022 but are not included on the WHO SDRM list, were detected in 12 (4.4%) and 9 (3.3%) samples, respectively. E138A was identified in six subtype B sequences and six subtype A1 sequences (Appendix A).

The majority of persons carrying SDRMs associated with any drug class were MSM infected with subtype B (66.7%, 30/45), while the others were MSM of subtype A1 (1/45), B-D recombinants (6/45), and CRF 39_BF (1/45). Five persons were infected with subtype B and one with CRF12_BF through heterosexual contact. One person carrying SDRM of unknown transmission risk was infected with subtype B.

Additionally, we explored the association between SDRMs and the demographic, epidemiological, and clinical characteristics of persons included in this study. Statistical analysis revealed that there were no significant associations between the SDRMs in the study population and age, transmission risk group, plasma viraemia, CD4+ T-cell count, gender, or subtype (Appendix A).

### 3.3. Deep Sequencing Analysis

A total of 60 persons were subjected to this part of the study where we compared patterns of SDRMs detected by SS and DS. To compare the results of the two sequencing methods, we considered a 5% and 15% threshold for DS and SS, respectively.

In 8/60 (13.3%) samples, no SDRMs were detected by any method. SDRMs were detected by DS in 52 samples (86.7%) and by SS in 20 samples (33.3%). A total of 31 and 72 SDRMs were identified by SS and DS, respectively. In detail, 49 SDRMs, including 44 NRTI-associated, 1 NNRTI-associated, 1 PI-associated, and 3 INSTI-associated SDRMs were identified by DS but missed by SS. Of those 49 SDRMs, 39 were identified at frequency <15%, and 7 of them were identified at frequency >15%. Although a threshold of 5% was considered for DS analysis, in one sample we reported a mutation with frequency <5% because it was identified by SS as well. These discrepancies in mutation frequencies are likely caused by the inherent stochasticity of the PCR process as well as the imperfect removal of the synthetic primer sequences from the raw reads.

Furthermore, eight SDRMs were detected only by SS and missed by DS, including two NRTI-associated, three NNRTI-associated, and three PI-associated SDRMs. The reason for these discrepancies is likely caused by filtering a portion of total NGS reads due to their poor mapping quality, which consequently resulted in insufficient coverage of the target region that contains SDRMs detected by SS.

Complete mutation concordance when both SS and DS identified the same pattern of SDRMs was observed in 8/52 (15.4%) samples. Partial concordance was identified in 7/52 (13.5%) samples where DS identified either additional mutations or SS identified mutations that were missed by DS. Complete divergence was observed in 34 samples (65.4%) when different patterns of SDRMs were detected by SS and DS or when only DS detected additional mutations. These discordant results mostly refer to samples where DS detected T215S as a single mutation or in combination with other mutations at frequencies <15% (28 samples) or >15% (5 samples) (Appendix A).

### 3.4. Phylogenetic Analysis

Phylogenetic analysis was performed on sequences of the most prevalent subtypes. In accordance with that, maximum likelihood trees were constructed separately for 191 sequences of subtype B, 28 B-D recombinant sequences, and 10 sequences of subtype A1. Phylogenetic analysis was not performed for the other subtypes, recombinants, or CRFs due to an insufficient number of sequences. The dataset for subtype B and B-D recombinants was supplemented with sequences from the period 2014–2017 and non-Croatian sequences as background sequences. That dataset enabled the identification of active as well as newly formed clusters. Namely, active clusters were defined as a group of sequences from the old (2014–2017) and new (2019–2022) periods, while the newly formed clusters consisted only of sequences from the new period (2019–2022).

For the analysis of cluster size, we defined clusters with 10 persons or more as large and clusters with less than 10 persons as small. Based on the defined settings for TCs, a total of 40 TCs with sequences in the present study were determined by phylogenetic analysis, precisely 30/40 (75.0%) of subtype B, 7/40 (17.5%) of B-D recombinants, and 3/40 (7.5%) of subtype A1. In addition, we identified 14 transmission pairs consisting only of sequences from the study period or in combination with a background sequence.

We described transmission clusters in relation to gender (male vs. female), age, stage of HIV diagnosis (acute vs. non-acute), plasma viraemia, CD4 count, transmission risk (MSM vs. non-MSM), HIV subtype (B vs. non-B), and SDRMs. For the majority of the selected demographic, epidemiological, clinical, and laboratory data, no statistically significant association between persons observed in TCs as compared to persons outside TCs was found, except for age (*p* > 0.001). Namely, individuals of a younger age were more likely to be identified in TCs (Table 2).

The subtype A1 phylogenetic tree revealed that local sequences were part of three mixed clusters. Namely, two mixed clusters containing one local sequence each, which indicates a separate introduction of these strains into the country, and one cluster containing six local sequences was identified for subtype A1 (Appendix A).

Phylogenetic analysis of the B-D recombinants revealed that 15 Croatian sequences from the present study were part of three newly formed local clusters (two local and one mixed). Another seven sequences from the present study were part of three local active clusters and one mixed cluster containing sequences from the period 2014–2017. Sequences with SDRMs T215S (*n* = 2) and I85V (*n* = 2) were part of two local clusters. In addition to that, two transmission pairs with Croatian and non-Croatian sequences with a support threshold of >0.9 were identified as well (Appendix A).

Phylogenetic data of subtype B accounted for the largest proportion of the analysed cohort and included 191 sequences from the present study (new dataset) and 536 background sequences. A total of 173/536 (32.3%) background sequences were Croatian sequences from the period 2014–2017 (old dataset).

Phylogenetic analysis for subtype B revealed that 74.9% (143/191) of sequences were a part of 16 local TCs with Croatian sequences (old and new datasets) and 14 mixed TCs with non-Croatian sequences with a support threshold of >0.9. In detail, we identified 8 TCs (26.7%) with up to 5 persons, 15 TCs (50.0%) with 5 to 15 persons, and 7 TCs (23.3%) with ≥16 persons. Overall, we identified 14 small clusters and 16 large clusters with sizes ranging from 3–45 persons. All identified clusters were active, with 18 TCs among them expanding throughout the study period, while 12 were newly formed. Among the newly formed TCs, three were local and nine included non-Croatian sequences. In addition, the phylogenetic analysis identified 12 transmission pairs with bootstrap support >0.9 containing Croatian sequences from the study period (Figure 1). Ten TCs with ≥5 Croatian sequences and their characteristics are presented in Table 3.

One sequence (accession number OR606232) seems to be unassigned, even though it has been classified as subtype B by the Rega HIV-1 Subtyping Tool, version 3.0.

Only one subtype B sequence carrying SDRMs was not included in the phylogenetic analysis. A total of 33 subtype B sequences carrying SDRM were part of the TCs. Forward transmission throughout the period 2019–2022 was determined in several clusters: (1) T215S (*n* = 19), with its sub-cluster T215S + L210W (*n* = 4), (2) K101E (*n* = 9), and (3) K103N (*n* = 5). One TC was newly formed, consisting only of sequences from present study and two of them were expanding clusters consisting of sequences from the new and old datasets.

A comparison of the clinical, demographic, and molecular characteristics of these TCs consisting only of sequences from the new dataset (2019–2022) is given in Table 4.

T215S remained the most common SDRM, determined in clusters as in the previous study. Namely, 45 sequences from the old and new datasets formed one large cluster with SDRM T215S, with a sub-cluster T215S + L210W expanding throughout the period of the new dataset. One sequence from the present study with SDRMs M41L + T215Y was identified in one local cluster consisting of sequences from the old and new datasets but they did not share that SDRM pattern.

One sequence with SDRM pattern L100I + K103N did not form a cluster with any of the sequences, indicating an independent route of infection with no evidence of transmission.

## 4. Discussion

In order to characterize the HIV epidemic in Croatia by analysing HIV-1 diversity, TDR, and transmission clusters, we combined virological, clinical, molecular, and epidemiological data from 270 newly diagnosed, treatment-naïve persons with HIV who enrolled in clinical care at the Reference Centre for HIV-AIDS at UHID in the period 2019–2022. More than 90% of treatment-naïve PLHIV that entered clinical care in the study period were included in this nationally representative study where we continued the longitudinal monitoring of TDR in Croatia.

The majority of the study population was men (94.1%) and the most prevalent age group was 18–35 years. Most of the patients originated from and reported Croatia as the country of infection and were included in the MSM risk group (94.8% and 85.6%, respectively). The characteristics of the study group are in agreement with the reports of our last nationwide study [31]. Our study cohort consisted of >50% of late presenters with a median CD4 count of 284 cells/μL and it is worth mentioning that a high prevalence of very late presenters (38.5%) was identified among them. According to the European Late Presenter Consensus Working Group, late presentation for HIV care refers to individuals presenting for care with a CD4 count below 350 cells/µL, or with an AIDS-defining event regardless of the CD4 cell count, while very late presenters (VLP) are characterized by presenting a CD4 count lower than 200 cells/μL or an AIDS-defining event, regardless of the CD4 cell count [48]. The data of two large European cohort studies in the period 2010–2016 and the ECDC data show that even in high-income countries such as in the European region, late presentation accounts for 36.9 to 64.2% of all HIV diagnoses [49,50].

In this study, we reported the prevalence of SDRMs in Croatia in the period 2019–2022. The prevalence of SDRMs was estimated at 16.7% over the study period. The HIV-1 epidemic in Croatia is characterized by a high prevalence of TDR, which was reported in our last two nationwide studies [22,31]. Although, a decreasing trend for TDR was observed over time (22% in the period 2006–2008 vs. 16.4% in the period 2014–2017), these numbers are still higher than the average of other European countries and even countries from the same region [23]. The European Society for Translational Antiviral Research (ESAR) is responsible for monitoring the prevalence of TDR in Europe through the SPREAD program, and according to its last study, which included a dataset from 2008–2010, the overall prevalence of TDR was estimated to be 8.3% [23]. The data on TDR have not been updated since 2016, but in a more recent study it was shown that during 2011–2019, the TDR prevalence in Europe increased to 12.8% [51]. TDR differs in different regions of the world and can change with the availability and introduction of new treatment regimens. Namely, the prevalence of TDR for the period 2014–2019 was 14.2% in North America, followed by 9.1% in Latin America and the Caribbean, 8.5% in Europe, 6.0% in sub-Saharan Africa, and 4.1% in southeast Asia [23,52]. In the Balkan region, the numbers for TDR vary from 14.8% reported in Romania, 8.8% in Serbia, and 7.8% in Greece to 2.4% in Slovenia [24,25,28,53].

The surveillance drug resistance list consisting of non-polymorphic HIV-1 DRMs to monitor TDR was generated by an expert panel in 2009 and was adopted by WHO and other institutions, and since then, it has been used to standardize monitoring of TDR in different countries [54,55]. In our study, TDR to NRTI was the most common (9.6%), with revertant T215S being the most frequent (8.1%) as in the study of Oroz et al. [31]. SDRMs T215C/D/S are revertants of the primary zidovudine (AZT) resistance mutations T215Y/F that have continued to spread in drug-naïve persons and among Croatian patients since the early 90 s [56].

In addition, we identified a rare mutation T69D in a male person infected with CRF 39_BF, for whom Brazil was reported as the likely country of infection. T69D is a non-polymorphic NRTI-selected mutation that reduces susceptibility to didanosine (ddI) and possibly d4T [57].

Resistance to NNRTIs has increased globally since 2013 in ART-naïve adults [52]. Consistent with the previous study, the prevalence of SDRMs to NNRTI remained fairly the same, observed in 20 persons (7.4%). The most frequent NNRTI was K103N, identified in 10 persons, followed by K101E, found in 9 persons. K103N is the most commonly transmitted nonpolymorphic NNRTI-drug-resistance mutation seen in people receiving nevirapine (NVP) and efavirenz (EFV), but when found in individuals before starting the WHO-recommended first-line regimen tenofovir (TDF)/lamivudine (3TC)/(EFV), it is associated with an increased risk of virological failure [58].

Based on the results of the ATLAS and FLAIR clinical trials, long-acting injectable cabotegravir and rilpivirine treatment (CAB/RPV LAI) has recently been approved for use in virologically-suppressed patients [59,60,61]. Since CAB/RPV LAI are currently being implemented across Europe, risk factors for confirmed virological failure, including HIV-1 subtypes A1/A6 as well as rilpivirine-associated resistance, should be carefully considered in various populations and geographic regions [60,62].

The 2022 update of the International Antiviral Society-USA (IAS-USA) on drug resistance mutations lists 16 mutations associated with decreased susceptibility to rilpivirine (RPV) (K101E/P, E138A/G/K/Q/R, V179L, Y181C/I/V, H221Y, F227C, M230I/L, and Y188L) [36]. In this study, we identified mutations E138A and E138K in 10 (3.7%) and 9 (3.3%) samples, respectively. Mutation E138A occurs as a natural polymorphism, especially in non-B subtypes [23], and in the present study, E138A was found in six subtype B and six subtype A1 sequences. E138K is a non-polymorphic mutation selected in a high proportion of persons receiving RPV, and in combination with K101E or the NRTI-resistance mutation M184I, it is sufficient to cause VF on a first-line RPV-containing regimen [63,64,65,66].

The HIV Drug Resistance Stanford database defines E138A as a polymorphic mutation weakly associated with reduced susceptibility to etravirine (ETR) and RPV [35], whereas the French HIV Resistance database (ANRS) defines E138A viruses as fully resistant to RPV [67]. E138A was not included in the surveillance drug resistance mutation (SDRMs) list recommended by the World Health Organization (WHO) for the surveillance of transmitted HIV drug resistance [54], but some researchers have taken this mutation into account when monitoring NNRTI-drug resistance in treatment-naïve HIV-1 patients [14,68,69,70]. The higher prevalence of NNRTI-associated E138A and K103N mutations is inconsistent with previous reports [71,72].

Fewer cases of SDRMs to PI were observed in the present study (1.5%) than in the previous (2.5%) study [31]. SDRMs to PI were identified in two persons with triple class resistance as well as in one person with single and double class resistance. In addition, we identified two persons with triple class resistance infected with subtype B and B-D recombinant with mutation patterns T215D, L100I, K103N, V32I, I47V, F53L and T215E, L100I, K103N, V32I, I47V, respectively.

INSTIs, especially second-generation inhibitors, are a favourable first-line regimen recommended for ART-naïve PLHIV by the ART guidelines [73]. TDR to INSTIs was first described in 2011 [74,75], and to this date, a very low prevalence of TDR to INSTIs was reported [76,77,78,79]. The data on TDR resistance to INSTIs are limited, although some studies reported the absence or very low level of TDR in the IN region [80,81,82,83]. The first large study on resistance to INSTIs was conducted by the European SPREAD study where the presence of INSTI-resistant variants in Europe (2006–2007) was analysed before the introduction of INSTIs. A few years back, surveillance of INSTI resistance was only recommended for patients with evidence of TDR to other drug classes and there was no obvious benefit of baseline TDR testing to INSTI in persons who entered clinical care [84]. However, the situation has changed over the years, with second-generation integrase inhibitors such as dolutegravir being recommended for first-line HIV treatment regimens following an increase in pre-treatment drug resistance to NNRTI-based regimens globally [85,86].

Resistance testing for INSTI was introduced in Croatia in 2015, and in the beginning, was performed only on selected patients. Today, resistance testing to INSTIs in Croatia is performed on all newly diagnosed HIV-infected persons. To this day, only two studies have been published on resistance to INSTI in Croatia, regarding treatment-naïve and experienced patients, respectively [31,87]. The present study is to date the largest cohort of ART-naïve persons with available resistance data on INSTI. The current study found no SDRMs in the IN region, which is in concordance with other European countries that reported a similar absence or very low levels of SDRMS to INSTIs [88,89,90,91,92,93]. Although no major INSTI mutations were found in this study, we identified a high prevalence of polymorphism S230N found in 50 patients (18.9%), which is not associated with reduced INSTI susceptibility. In addition, the integrase resistance associated accessory mutation L74I was found in seven patients and was more common in non-B subtypes. In the Stanford Resistance Database [35], L74I is reported in combination with other integrase mutations with a prevalence of 3% to 20%, depending on subtype, but it is not listed on the IAS-USA drug resistance mutations list [36]. L74I has a minimal effect on INSTI susceptibility but can contribute to reduced susceptibility to each of the INSTIs when it occurs in association with primary INSTIs mutations [94,95,96].

Although no prevalence of resistance to integrase inhibitors was observed, as expected, the study population consisted of a high proportion of late presenters, so the level of SDRMs to INSTI could be underestimated due to a loss of DRM-possessing viruses in an actively replicating viral pool, as suggested in a previous study [97].

Our sequence analysis confirmed that the majority of infections (73.3%) were classified as subtype B, which is common in most western European countries [23] and countries of the Balkan region [98], except for Romania and Albania, where the most prevalent subtypes are F1 (80.3%) and A1 (56.1%) [99,100]. In previous nationwide studies reported by Oroz et al. [31] and Grgic et al. [22], subtype B infection accounted for 91.3% and 89.0% of all new HIV infections, respectively, so there was a slight decline of subtype B in favour of non-B subtype infections in recent years. Subtype A1, subtype C, subtype F, CRF12_BF, and CRF01_AE were observed in >1% of the cohort. We also reported a significant number of different recombinant forms. In the present study, 42 sequences were classified as recombinants, and the majority of them were recombinants of subtype B and D identified in 29 individuals. In order to explore the transmission network of those sequences, we included 28 sequences of B-D recombinants in a separate phylogenetic analysis. Given that the majority of our cohort consisted of subtype B sequences, our phylogenetic analysis mainly focused on the subtype B transmission network.

Phylogenetic interference for subtype B sequences that consisted of new (2019–2022) and old (2014–2017) datasets as well as background sequences revealed 30 TCs (16 local TCs with Croatian sequences from the old and new datasets and 14 mixed TCs with non-Croatian sequences). All clusters were active, and newly formed clusters (53.3%) were somewhat more represented than the expanding clusters (46.7%). Among the newly formed TCs, 25% were local and 75% were mixed clusters, indicating a cross-border transmission during the study period regardless of the ongoing COVID-19 pandemic. The recent trend of increased risk factors, such as travelling abroad for chem-sex parties, is linked to cross-border transmission and was reported by some studies [101,102]. Since the origin of the background sequences was not available we did not explore the geographic relationships between the clustered sequences.

Phylogenetic analysis revealed that a total of 143/191 (74.9%) persons were part of TCs, of which >50% were found in local TCs. The findings in our study are comparable with studies from other western and central European countries, where HIV-1 infection is concentrated among MSM infected with subtype B, but the proportion of clustered sequences was <60% [4,103,104,105,106,107]. Less clustering was observed in the present study than in the previous (74.9% vs. 86%), but it should be taken into consideration that fewer sequences were subjected to the TC analysis than in the previous study due to fewer subtype B infections.

A total of 39/45 (86.7%) persons carrying SDRMs were observed in TCs with forward transmission confirmed in 4 local clusters. We identified one large, expanding T215S cluster with its sub-cluster T215S + L210W, one expanding K101E cluster, and one newly formed K103N cluster. The previous study identified a T215S cluster of 53 persons as the largest local SDRM cluster in Croatia [31]. Phylodynamic analysis suggested that this TC could be one of the earliest clusters in the country, with an estimated tMRCA in the year 1992. The largest local cluster identified in the present study was also T215S, expanding throughout the study period. T215 revertants are one of the most common NRTI mutations found in treatment-naïve persons [23,108] and the high prevalence of T215S revertants in the Croatian HIV-1 cohort is likely due to this once highly prevalent therapy in the country. Along with T215S, we found one local K103N cluster where all people participating were MSM. Another rather large TC identified in this study was the K101E cluster consisting of 21 sequences of different transmission risk groups from the old and new datasets.

In the previous nationwide study, Oroz et al. identified one local TC with a triple class resistance pattern with an origin back in 2008 [31]. In the present study, we identified two persons with triple class resistance with a similar pattern (T215D, L100I, K103N, V32I, I47V, F53L/T215E, L100I, K103N, V32I, I47V), but since they were not part of the phylogenetic analysis, possible transmission was not determined.

When combining the socio-demographic data of persons included in this analysis with the results of the phylogenetic inference, we found that transmission in clusters in Croatia involved young MSM infected with subtype B, which highlights the importance of continued education on HIV prevention among that risk group and other persons at risk in order to reduce the risk of infection and minimize the transmission of the virus.

There are several explanations for the high TDR reported in our study. Firstly, the comparison of prevalences between different studies is challenging due to the different algorithms used to identify TDR mutations. The WHO list for TDR surveillance has not been updated since 2009 and tends to overestimate the importance of polymorphisms, revertant mutations, and mutations only affecting the activity of drugs that are no longer prescribed, and it does not take into account some important mutations that implicate currently used ARV [109]. TDR is significantly lower (7.8%) when a Stanford mutation score is set at ≥60 to identify the mutations with a high impact on drug susceptibility. Secondly, the main reason for the high TDR reported in our study is the large TC of MSM with SDRM T215S, which according to Oroz et al. dates back to 1992, and the relatively common K101E cluster, which dates back to 2008 [31]. It should be emphasized that the current resistance profiles do not compromise the current therapy, that is, the drugs that are used in treatment today.

To evaluate DS for identifying low abundance variants and to compare the results with SS, we randomly selected 60 samples. This is the largest comparison between these two methods conducted in Croatia so far. HIV-1 minority variants exist in <15% of the entire HIV-1 viral population [110] and DRMs in HIV-1 minority variants could compromise the effectiveness of ART treatment, especially for NNRTI-based regimens [111]. A significant number of resistant variants in initial infection can pass under the radar of the standard level of detection (>15%), especially in late presenters [112,113]. DS identified an additional 49 SDRMs and the most frequent SDRMs identified by DS and missed by SS was T215S. Complete and partial concordance between the two methods was observed in 15.4% and 13.5%, respectively. In the majority of samples (65.4%), SS and DS disagreed but the discrepancies mostly referred to samples where DS detected T215S as an additional mutation. Our results confirmed that disagreement between the two platforms is possible, as was reported by other studies [114]. The major limitations of our DS approach are the stochasticity of PCR itself; an additional step of primer removal that can introduce errors, i.e., differences in frequencies and mask alternative bases that complicate the analysis; and the heterogeneous amplification efficiency of small PCR fragments that results in uneven coverage and a potential lack of minimum coverage. Regardless of these problems, our results show the value of this approach and that it is obtaining additional information.

## 5. Conclusions

The results of our study give insight into the transmission dynamics of an HIV epidemic in Croatia that is locally driven by young MSM infected with subtype B. We confirmed a high prevalence of SDRMs of 16.7% as well as forward transmission of resistance strains. Besides TDR associated with NRTIs, the emergence of SDRMs to NNRTIs as well as the higher incidence of non-B subtypes observed in this study suggest important changes in the pattern of primary resistance and viral diversity. The majority of identified mutations were clinically relevant, which makes baseline HIV resistance testing in Croatia of unquestionable importance. Although analysis of the clinical significance of SDRMs showed a favourable virological background for INSTI-based treatment since no SDRMs to INSTI were observed, surveillance of TDR to INSTI should be continued in terms of ascertaining the role of polymorphic and non-polymorphic mutations in the development of resistance. In addition, we explored the performance of DS in detecting minority HIV-1 variants and demonstrated a valuable tool for HIV drug resistance assessment with an emphasis on the importance of standardization between the two methods, especially in terms of the very well-established threshold for drug resistance. Our findings demonstrate the importance of continuing to monitor SDRMs and transmission clusters in Croatia in order to facilitate enhanced HIV prevention efforts.

## Figures and Tables

**Figure 1 viruses-15-02408-f001:**
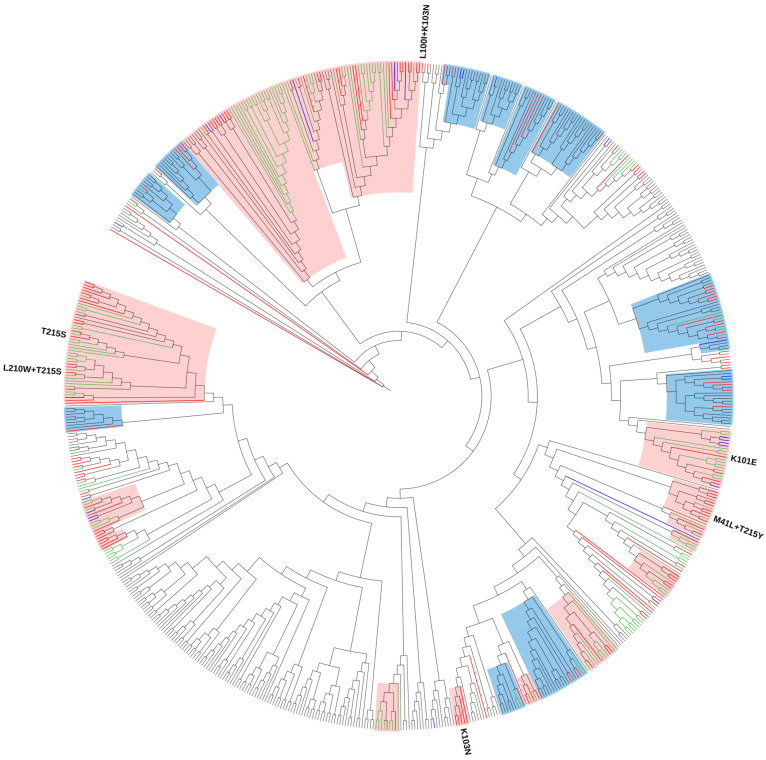
Maximum likelihood phylogenetic analysis of the Croatian HIV-1 subtype B sequences from the old (2014–2017) and new (2019–2022) datasets and background sequences. Branches of Croatian sequences from the new dataset are coloured according to the transmission risk: (MSM: men who have sex with men—red; heterosexual—blue). Branches of sequences from the old dataset are coloured green, while branches of all background sequences are coloured black. All identified surveillance drug resistance mutations (SDRMs) are positioned on the tree along with the corresponding sequences. For SDRMs T215S, T215S  +  L210W, K101E, and K103N that formed local transmission clusters (TCs), above each cluster, the corresponding SDRMs are noted. TCs with >75% of Croatian sequences (local clusters) are highlighted in light pink, while TCs with <75% of Croatian sequences (mixed clusters) are highlighted in light blue.

**Table 1 viruses-15-02408-t001:** Characteristics of newly diagnosed people living with HIV-1 (PLHIV) included in the study period 2019–2022.

Patient Characteristics	*N*	%
**Total**	270	100.0
Men	254	94.1
Women	16	5.9
**Age at HIV diagnosis (years)**		
18–35	111	41.1
36–47	95	35.2
48–59	53	19.6
>60	11	4.1
**Median age (Q1–Q3), years**	38.0 (29.1–46.6)	
**Reported country of infection**		
Croatia	256	94.8
Other	13	4.8
Unknown	1	0.4
**Stage at HIV diagnosis**		
Acute infection	114	42.2
Chronic infection (late presenters)	51	18.9
Chronic infection (very late presenters)	104	38.5
Unknown	1	0.4
**Transmission**		
MSM	231	85.6
Heterosexual	32	11.9
IDU	1	0.4
Unknown	6	2.2
**Year of inclusion in clinical care**		
2019	83	30.7
2020	53	19.6
2021	67	24.8
2022	67	24.8
**Baseline CD4+T cells/μL ***		
<100	68	25.2
101–250	54	20.0
251–400	63	23.3
401–600	44	16.3
>601	40	14.8
**Baseline CD4+T cells/μL, median (Q1–Q3)**	284 (95–460)	
**Log10 baseline plasma viraemia, median** **(Q1–Q3)**	5.0 (4.4–5.5)	

MSM: men who have sex with men; IDU: injecting drug users; Q1, Q3: first and third quartile; *N*: number of individuals. * CD4+ was unavailable for one person.

**Table 2 viruses-15-02408-t002:** Comparison of clinical, socio-demographic, and virological characteristics of people living with HIV-1 (PLHIV) in the period 2019–2022 included in the phylogenetic analysis.

	Total, *N* (%)	Persons in TC	Persons Outside TC	*p*-Values	Persons with SDRM	Persons without SDRM	*p*-Values
Persons, *n* (%)	229 (100.0)	173 (75.5)	56 (24.5)		39 (17.0)	190 (83.0)	
**Gender, *n* ^a^ (%)**				0.525			0.468
Male	216 (94.3)	164 (94.8)	52 (92.9)		36 (92.3)	180 (94.7)	
Female	13 (5.7)	9 (5.2)	4 (7.1)		3 (7.7)	10 (5.3)	
**Age at HIV diagnosis, median years (Q1–Q3) ^c^**	38.1 (29.0–47.0)	36.8 (28.2–46.3)	43.4 (36.5–47.5)	<0.001	33.1 (27.8–49.9)	38.9 (29.3–47.0)	0.359
**Stage at HIV diagnosis ^a^**				0.333			0.806
Acute infection	98 (42.8)	82 (47.4)	16 (28.6)		16 (42.1)	82 (43.2)	
Chronic infection (late presenters)	44 (19.2)	29 (16.8)	15 (26.8)		6 (15.8)	38 (20.0)	
Chronic infection (very late presenters)	86 (37.6)	61 (35.3)	25 (44.6)		16 (13.2)	70 (36.8)	
Unknown *	1 (0.4)	1 (0.6)	/		1 (2.6)	/	
**Log10 baseline plasma viraemia, median (Q1–Q3) ^c^**	4.9 (4.4–5.5)	4.9 (4.4–5.5)	5.0 (4.4–5.5)	0.561	4.8 (4.4–5.4)	5.0 (4.4–5.6)	0.374
**Baseline CD4+ T cells/μL, median (Q1–Q3) ^c^**	300.5 (95.0–460.0)	338.0 (118.5–497.5)	243.0 (43.5–374.0)	0.048	310.5 (87.0–447.0)	296.5 (95.0–468.0)	0.695
**Transmission risk ^b^**				0.756			0.407
MSM	203 (88.6)	154 (89.0)	49 (87.5)		33 (84.6)	170 (89.5)	
Heterosexual	20 (8.7)	15 (8.7)	5 (8.9)		5 (12.8)	15 (7.9)	
IDU *	/	/	/		/	/	
Unknown *	6 (2.6)	4 (2.3)	2 (3.6)		1 (2.6)	5 (2.6)	
**HIV subtype, *n* (%) ^b^**				0.593			0.599
B	191 (83.4)	143 (82.7)	48 (85.7)		33 (84.6)	158 (83.2)	
A1	10 (4.4)	8 (4.6)	2 (3.6)		1 (2.6)	9 (4.7)	
BD	28 (12.2)	22 (12.7)	6 (10.7)		5 (12.8)	23 (12.1)	

TC: transmission cluster; SDRMs: surveillance drug resistance mutations; MSM: men who have sex with men; IDU: injecting drug users; Q1, Q3: first and third quartile; *N*: number of individuals; CD4+ was unavailable for one person; ^a^: associations for categorical variables were tested using Chi-squared tests; ^b^: associations for categorical variables were tested using Fisher’s exact test; ^c^: associations for continuous variables were tested using Mann–Whitney tests; * variables with a small number of samples were excluded from the sum of proportions and the statistical analysis.

**Table 3 viruses-15-02408-t003:** Characteristics of transmission clusters comprising ≥5 Croatian sequences with bootstrap threshold >0.90 identified in the phylogenetic analysis.

Cluster, Total Number of Sequences, *n*	Croatian Sequences, *n*	Bootstrap Threshold	Type of Cluster	SDRMs	Subtype
1 (45)	45	0.92	Expanding	T215S	B
2 (41)	33	0.97	Expanding		B
3 (15)	15	0.97	Expanding		B
4 (36)	32	0.97	Expanding		B
5 (24)	15	0.93	Expanding		B
6 (21)	21	0.98	Expanding		B
7 (9)	9	0.90	Expanding		B
8 (13)	13	0.98	Expanding		B
9 (5)	5	0.98	Newly formed	K103N	B
10 (12)	12	0.96	Expanding		B

SDRMs: Surveillance drug resistance mutations.

**Table 4 viruses-15-02408-t004:** Main characteristics of Croatian transmission clusters harbouring SDRMs.

	T215S Cluster	T215S + L210W Cluster	K101E Cluster	K103N Cluster
Total number of individuals, *n* (%)	19 (100.0)	4 (100.0)	9 (100.0)	5 (100.0)
Gender			6 (66.7)	
Male	19 (100.00)	4 (100.0)	3 (33.3)	5 (100.0)
Female	/	/		
Transmission risk				
MSM	18 (94.7)	4 (100.0)	4 (44.4)	5 (100.0)
Hetero	/	/	5 (55.5)	
Unknown	1 (5.3)	/	/	
Reported country of infection				
Croatia	19 (100.0)	4 (100.0)	9 (100.0)	5 (100.0)
Other	/			/
Stage at HIV diagnosis				
Acute infection	7	1 (25.0)	/	4 (80.0)
Chronic infection (late presenters)	4	1 (25.0)	2 (22.2)	1 (20.0)
Chronic infection (very late presenters)	7	2 (50.0)	7 (77.8)	/
Unknown	1	/	/	/
Age, median (Q1–Q3), years	32.0 (25.0–38.6)	35.2(27.6–39.7)	49.9 (33.0–54.4)	50.8 (48.8–59.2)

SDRMs: Surveillance drug resistance mutations; MSM: men who have sex with men; Q1, Q3: first and third quartile; *N*: number of individuals. CD4+ was unavailable for one person.

## Data Availability

All data analysed in this study are included in this published article and its Appendix A. All sequences obtained in this study by SS were submitted to the GeneBank nucleotide sequence database under accession numbers OR605755-OR606279 and OR613869-OR613875. Sequences generated by DS are available from the ArrayExpress database at EMBL-EBI under accession number E-MTAB-13563.

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
