# Peer review of "Characterization of Human Immunodeficiency Virus-1 Transmission Clusters and Transmitted Drug-Resistant Mutations in Croatia from 2019 to 2022"

_viruses, 2023, doi:10.3390/v15122408_

Round 1
Reviewer 1 Report
Comments and Suggestions for Authors
The study focuses on the molecular epidemiology and HIV drug resistance in Croatia, using viral samples from 270 treatment-naive individuals infected with HIV-1 between 2019 and 2022.
The estimated prevalence of primary HIV drug resistance is 16.7%, with the majority of identified mutations being clinically relevant. The most vulnerable group comprises MSM aged 18-46 years, and Subtype B is the most common (73.3%). Comparison of deep sequencing with Sanger sequencing revealed the presence of additional surveillance drug resistance mutations in most of the samples studied. Cluster analysis demonstrated the transmission of virus variants resistant to antiretroviral drugs, with phylogenetic clusters predominantly composed of MSM.
The data obtained underscore the necessity of further monitoring of HIV-1 infection epidemiology, including transmission networks.
The study employs detailed and modern analysis methods, and the conclusions align with the results obtained. However, the Discussion section is lengthy and contains numerous repetitions and unnecessary explanations, suggesting a need for condensation. Additionally, there are some typos and not entirely successful expressions.
Some easily corrected comments are listed below.
1. Line 76. “TDR resistance in newly diagnosed HIV-infected patients”. Traditionally, transmitted drug resistance is analyzed in recently infected persons. The terms “primary resistance” and “treatment-naïve” are more relevant in this context.
2. Line 77. “showed a high prevalence of subtype B”. The percentage should be indicated.
3. Lines 88-97. It should be noted that the vast majority of patients had mutations to only one class of drugs (as far as is clear from the text, there were no cases of resistance to two classes). You should also indicate the number of patients in each group, not just the percentage.
4. Line 98. “there was a shift in transmission routes in Croatia”. It is not clear from the text which route of transmission of the virus predominated before the shift occurred.
5. Line 474. “a decreasing trend for TDR was observed over time”. The reliability of the trend should be calculated.
Comments on the Quality of English Language
no specail comments
Author Response
Responses to reviewers
Reviewer 1:
Comment:
1.Line 76. “TDR resistance in newly diagnosed HIV-infected patients”. Traditionally, transmitted drug resistance is analyzed in recently infected persons. The terms “primary resistance” and “treatment-naive” are more relevant in this context.
Answer: The correction is highlighted in yellow
A study on primary resistance in treatment-naïve HIV-infected patients conducted from 2006-2008 showed a high prevalence of subtype B (89%) among men who have sex with men (MSM) with only 11% of patients infected with non-B subtypes (Line 76 highlighted in yellow)
Comment:
2.Line 77. “showed a high prevalence of subtype B”. The percentage should be indicated.
Answer: The percentage is shown (89%) and highlighted in yellow (Line 77)
Comment:
3.Lines 88-97. It should be noted that the vast majority of patients had mutations to only one class of drugs (as far as is clear from the text, there were no cases of resistance to two classes). You should also indicate the number of patients in each group, not just the percentage.
Answer: The corrections are indicated in yellow (Line 90-97)
The majority of patients had mutations to only one class of drugs. Resistance to nucleoside reverse transcriptase inhibitors (NRTIs), non-nucleoside RTI (NNRTIs) and protease inhibitors (PIs) was found in 11.4% (46/403), 6.7% (27/403) and 2.5% (10/403) of persons, respectively with the addition of a single case (1.0%, 1/100) of resistance to integrase strand-transfer inhibitors (INSTIs). In nine individuals (2.2%) triple-class resistance was detected. Phylogenetic analysis showed that 86.1% of sequence where a part of transmission clusters and identified forward transmission resistance. The majority of infected people were of subtype B (91.3%, 368/403).
Comment:
4.Line 98. “there was a shift in transmission routes in Croatia”.It is not clear from the text which route of transmission of the virus predominated before the shift occurred.
Answer: The corrections are indicated in yellow (Line 99-102)
In the last two decades, there was a shift in transmission routes in Croatia in a way that the majority of HIV infections in the country now are acquired through MSM contact compared to the period 1985-2010 when 40–50% of new HIV-1 cases were attributed to MSM transmission
Comment:
5.Line 474. “a decreasing trend for TDR was observed over time”. The reliability of the trend should be calculated.
Answer: The corrections are highlighted in yellow
Although, a decreasing trend for TDR was observed over time (22% in the period 2006-2008 vs 16.4% in the period 2014-2017), the numbers are still higher than the average of other European countries and even countries from the region (Line 481)
The manuscript was run through spelling /grammar checker and the typos were corrected accordingly and are highlighted in orange.
As suggested by Reviewer 1, in order to condensate the section Discussion some changes were made. Everything what is considerate repetition and unnecessary is crossed out and colored in red and it should be removed once the changes are excepted by the reviewers.
One new reference was added (reference number 111) and some references were excluded (110-114 in original manuscript) and in accordance with that the order of references was corrected which is indicated in the manuscript.

Reviewer 2 Report
Comments and Suggestions for Authors
Very comprehensive surveillance, albeit on relatively few samples, 2019-2022.
- nothing methodologically new but useful results for this interested in this HIV population
- the authors need to try to explain more why their TR incidence is so high, E.g. incomplete viral suppression due to poor compliance, lost to followup, patients swapping drugs, etc. otherwise, the appropriate interventions cannot be developed and implemented
- run the text through spelling/grammar checker to remove a few typos
Otherwise a very nicely presented study.
Author Response
Responses to reviewers
Reviewer 2:
Comment 1:
The authors need to try to explain more why their TR incidence is so high, E.g. incomplete viral suppression due to poor compliance, lost to followup, patients swapping drugs, etc. otherwise, the appropriate interventions cannot be developed and implemented.
Answer: The explanation about why the TDR in our study is high is added and indicated in yellow (Lines 678-687)
There are several explanations for the high TDR reported in our study. Firstly, the comparison of prevalence rates between different studies is challenging due to the different algorithms used to identify TDR mutations. WHO list for TDR surveillance hasn't been updated since 2009 and tends to overestimate the importance of polymorphisms, revertant mutations and mutations only affecting the activity of drugs that are no longer perscribed but doesn’t take into account some important mutations that implicate currently used ARV [111]. TDR is significantly lower (7.8%) when a Stanford mutation score is set at ≥ 60 to identify the mutations with a high impact on drug susceptibility. Secondly, the main reason for the high TDR reported in our study is the large TC of MSM with SDRM T215S, which according to Oroz et al dates back to 1992 and the relatively common K101E cluster which dates back to 2008 [31]. It should be emphasized that the current resistance profiles do not compromise the current therapy, that is, the drugs that are used in treatment today.
Comment 2:
- run the text through spelling/grammar checker to remove a few typos
Answer: The manuscript was run through spelling /grammar checker and the typos were corrected accordingly and are highlighted in orange.
One new reference was added (reference number 111) and some references were excluded (110-114 in original manuscript) and in accordance with that the order of references was corrected which is indicated in the manuscript.
